# Bridging Neuroscience and Robotics: Spiking Neural Networks in Action

**DOI:** 10.3390/s23218880

**Published:** 2023-11-01

**Authors:** Alexander Jones, Vaibhav Gandhi, Adam Y. Mahiddine, Christian Huyck

**Affiliations:** Faculty of Science and Technology, Middlesex University, London NW4 4BT, UK; v.gandhi@mdx.ac.uk (V.G.); adam.mahiddine@gmail.com (A.Y.M.); c.huyck@mdx.ac.uk (C.H.)

**Keywords:** cell assemblies, lateralized readiness potential, LRP, spiking neural network, robot

## Abstract

Robots are becoming increasingly sophisticated in the execution of complex tasks. However, an area that requires development is the ability to act in dynamically changing environments. To advance this, developments have turned towards understanding the human brain and applying this to improve robotics. The present study used electroencephalogram (EEG) data recorded from 54 human participants whilst they performed a two-choice task. A build-up of motor activity starting around 400 ms before response onset, also known as the lateralized readiness potential (LRP), was observed. This indicates that actions are not simply binary processes but rather, response-preparation is gradual and occurs in a temporal window that can interact with the environment. In parallel, a robot arm executing a pick-and-place task was developed. The understanding from the EEG data and the robot arm were integrated into the final system, which included cell assemblies (CAs)—a simulated spiking neural network—to inform the robot to place the object left or right. Results showed that the neural data from the robot simulation were largely consistent with the human data. This neurorobotics study provides an example of how to integrate human brain recordings with simulated neural networks in order to drive a robot.

## 1. Introduction

Robots and robotic systems have been applied to a wide range of industrial applications and services. Robots are able to accomplish ever more complex tasks with incomparable meticulousness, for example neurosurgery [1]. Stability, dexterity, and accuracy are enhanced with the integration of advanced robots and skilled surgeons in neurosurgery when compared to traditional (manual) procedures [2]. However, complex decision making and judgements still rely heavily on human operators rather than purely on robots [2]. Moreover, there are technical bottlenecks that still exist for robots, such as moving with true flexibility, dynamically interacting with the environment, and having limited intelligent perception and control [3]. To advance the development of robots, interest has turned to arguably the most complex system and known processor, namely the human brain. The area of neurorobotics aims to capitalise on the understanding and development of neuroscience to improve the development of robots. Robots have been made to be more human-like via two main routes. The first is to mimic the way humans look and behave, often to complete the work humans do. These humanoid robots can perform varying human-like actions, such as mimicking the behaviours of a child [4] and even playing football [5].

The second route, which is more relevant to the current study, is to mimic the underlying and fundamental mechanisms of humans in order to develop brain-inspired intelligent robots. These human brain-inspired robots aim to mimic neural mechanisms during cognitive processing, such as decision making, sensory processing, and action control [6]. For example, Tang, Huang, Narayanamoorthy, and Yan [7] used the understanding of how the hippocampus and entorhinal cortex are necessary for spatial cognition in humans to develop a system to enable a mobile robot to perform task-based navigation in a maze environment. One of the goals of the Human Brain Project was to integrate research in human neural behaviour with robot development, for example to improve robot control [8]. Moreover, Oliver, Lanillos, and Cheng [9] used the understanding of how the brain encodes uncertainty and uses the difference between what is observed in the environment and what is expected to learn and perform actions using the free energy principle [10]. They used this prediction error (discrepancy between the visual and joint sensors) and its expected values to improve the robot’s ability to alter its body position during a task. A cornerstone of this approach is the creation of more flexible and efficient robots that are capable of learning and operating in a noisy and unpredictable environment [11].

Effectively interacting with a dynamic environment requires a balance between activating and inhibiting responses. Crossing the road involves activating an action and taking a step; however, if a car appears in the periphery, inhibition is needed to stop the action and avoid getting hit by the car. In the area of psychology and neuroscience, human action has been studied extensively using electroencephalography (EEG), which enables brain signals to be mapped into the digital domain (see, e.g., [12]). For example, when performing an action with one hand (or foot) there is a build-up of neural activity over contralateral motor areas before the response is emitted. This is known as the lateralized readiness potential (LRP) [13]. In general terms, the LRP is considered a measure of response activation or preparation. The LRP is typically best observed over electrodes C3/4 on the human scalp, contralateral to the hand performing the action, and is a negative deflecting wave, as measured with EEG. It is assumed that the motor cortex plays a large role in the generation of the LRP [13,14], in particular the premotor cortex [15]. The build-up of motor activity starts around 400 ms before movement onset, and the excitability of premotor areas increases more rapidly, showing a steeper negative slope, as the response approaches. The pre-movement build-up of activity is not unique to humans but has been observed in monkeys [16], rodents [17], and even fish [18].

In humans, the LRP has been extensively studied in different paradigms, such as performing an action or not (e.g., Go/No-Go task: [15,19] or selecting one of two possible responses (two-choice task) or the stop-signal task [20,21]. The stop-signal task has been primarily used as a measure of the inhibition of a planned action, as a result of a stop signal. On each trial, participants initiate and plan a movement, such as a keyboard press, in response to a go-signal. However, on a proportion of trials, a stop signal is presented shortly after the go signal, and the task for the participant is to simply cancel their response and not to perform the movement. In general, the longer the delay between the go signal and the stop signal, the harder it is to cancel the action. The more activation in the build-up of premotor response preparation, the harder it is to stop an action [22]. For example, Ko, Cheng, and Juan [23] measured the LRP activity in the 200 ms time interval before a stop signal and compared trials in which participants were able to successfully inhibit their actions with those in which they incorrectly made a response. The authors showed that a greater negative amplitude was associated with the times when participants were unable to cancel their actions. That is, when there is a greater build-up of neural activity before the action, a subject will be quicker to perform the action, but also less able to stop a response. This observed effect between successful or unsuccessful stopping of a planned action in a stop-signal task has been explained through the independent horse-race model [24]. According to the independent horse-race model, stopping success is determined by the relative finishing time of two commands, namely the stop signal or the go signal. When the go signal finishes first, the action is executed, but when the stop signal finishes first, the action is inhibited. The threshold between the two processes, whether to successfully stop an action or not, can be influenced by a range of different variables, such as the delay between commands, task familiarity, response conflict, probability of two actions, and so on (see [25] for a review). The important take-home message is that the actions humans perform are not binary but depend on the variable activity prior to the action. The process is adapted to support flexible and goal-directed behaviour in ever-changing environments. The current study uses an understanding of motor preparation from human psychology and neuroscience. To be more specific, EEG data from forty-seven participants planning to move their left or right hand were recorded. This established the presence of an LRP and showed differences in neural activity based on the decisions made. With this information, this neural response activity in humans can be used with greater confidence as the basis of a robot performing an action. This neuroscience-inspired robotics project also includes a simulated spiking neural network, which specifically uses a cell assembly (CA), to bridge the gap between human and robot action.

A long-standing hypothesis [26] states that the neural basis of concepts are recurrently connected sets of neurons that can maintain activation without external stimulus: CAs. There is significant evidence of CAs in biological brains (see [27,28] for reviews). A CA is both a long-term and a short-term memory. As the neurons in the CA start to fire, they spread the activation to other neurons within the CA that have strong connections from those firing neurons. Those neurons in turn activate other neurons, perhaps including the initial neurons, igniting the CA. Without sufficient synaptic strength, learned from earlier repeated activation and Hebbian learning, the CA would not exist. The long-term memory forms the CA, which can then be used as a short-term memory. Biologically, this is not a well-understood process, but there is some evidence supporting it, not least the evidence supporting the Hebbian spike timing-dependent plasticity learning rule (e.g., [29]). It is part of Hebb’s original cell assembly hypothesis.

A reasonable assumption is that the LRP is based on the activation of a particular CA. Initial neurons in the CA are stimulated by a signal. This then leads to a cascade of neural firing, and a growing level of firing within the CA. When the firing surpasses a threshold, the action is signalled. This, of course, is all done neurally—that is, the signal is sent on to other CAs, which then perform the action of, for instance, moving an arm.

The aim of this study is to take a multidisciplinary approach and integrate cognitive neuroscience and robotics. Specifically, it uses simulated cell assemblies to drive a robot action based on recorded data of motor build-up of neural activity in the human brain. In particular, a signal is sent to the CA, which starts the ignition process. When the ignition process has been completed, the action can no longer be stopped.

## 2. Materials and Methods

This study, together with the associated methods, is divided into three parts: (i) the recording and analysing of human EEG data to establish the presence of motor build-up activity in the brain prior to an action; (ii) the development of a robot to perform a pick-and-place task; and (iii) the use of cell assemblies to integrate the decision-making process into the robot. The three parts integrate by using the theory of the observed human neural activity in response to an action, then applying this pattern to a simulated spiking neural network, which in turn is implemented as part of a robot performing a task. To be more specific, EEG data were first recorded to establish the underlying neural output by which a human performs an action. Based upon this observed gradual build-up of motor activity in the human brain, a simulated spiking neural network was developed. A robot was then developed to establish whether the simulated spiking neural network—which was inspired by our human brain data—could then be implemented as part of the robot programme.

### 2.1. Human Electroencephalogram (EEG) Recording

Data were recorded from 54 participants aged between 18 and 35 years (M age = 23.44 years; SD = 4.84; 21 male) whilst they performed a memory task. Ethical approval was granted by Middlesex University Research Ethics Committee, and participants provided written informed consent. After pre-processing of the EEG data (see below) 47 participants were included in the LRP analysis. Here, an overview of the materials, methods, and procedures relevant to the present project are provided. However, further details can also be found in Jones, Silas, Anderson, and Ward [30].

#### Task and Procedure

Participants were seated in a sound attenuated room in front of a PC monitor and EEG was recorded throughout the experiment. Participants engaged in an encoding task followed by a memory task (see Figure 1 for a schematic view of the procedure). This was repeated eight times.

In total, there were 557 greyscale images of objects taken from Jones and Ward [31]. Each encoding phase contained a unique set of 30 critical items randomly interspersed among 90 presentations of a checkerboard. The participants’ task during the encoding phase was to press the space bar every time the image was an animal. Pictures of an animal made up about 10% of the object items. Animal items were included to make sure participants focused their attention and processed the items fully but were not analyzed further. The encoding task took about 5 min to complete and was followed by a recognition task. Each recognition test phase contained the 30 objects from the immediately prior encoding phase, along with 30 completely new objects. In other words, participants were presented with 60 images and had to respond whether they had seen the object before or not. Each recognition task took about 6 min to complete. The current study only investigated the build-up of motor activity during the yes/no response in the recognition task.

### 2.2. EEG Recording and Pre-Processing

EEG data were recorded from 64 locations on the scalp throughout the experiment with a sample rate of 1000 Hz using a Brain Products ActiChamp system. Horizontal electro-oculogram (HEOG) data were recorded from the outer canthi of the eyes. Offline data analysis (Brain Vision Analyzer, Brain Products GmbH, Gilching, Germany) included interpolation of bad channels, identified manually, on a participant-by-participant basis. A second-order Butterworth zero-phase band-pass filter with low cut-off of 0.1 Hz and a high cut-off of 40 Hz, and a 50 Hz zero-phase notch filter were applied to each participant’s continuous data, and data were re-referenced to the average of all 64 electrodes [32]. Eye blinks were corrected in a semi-automatic mode, using ocular correction independent component analysis (ICA). Data were again high-pass filtered at 0.5 Hz, removing the need for baseline correction and suppressing DC voltage fluctuations (see, e.g., [33]). For the LRP analysis, ERPs were segmented into 1700 ms intervals separately for correctly identified old items (yes responses) and correctly identified new items (no responses). The segment interval was between 1500 ms and 200 ms around action onset (0 ms). Artefact rejection was performed on all channels excluding segments with amplitudes greater than ±100 µV. To calculate the LRP, ipsilateral brain activity over motor areas was subtracted from contralateral activity. That is, for right-hand responses (yes responses/remembered old items) the average amplitude for electrode C4 was subtracted from C3. For left-hand responses (no responses/new items), the calculation was C4-C3 [34]).

### 2.3. Robot Development

The robot arm was designed and assembled for a 3 DOF flexibility and dexterity to be able to perform a pick-and-place task (see Figure 2). The 3 DOF involves an ability to move up or down, sideways, and forward or backward. The robot consists of AX-12a servo motors, a custom-made gripper and linear actuator, force-sensitive resistors (FSRs), an Arduino-compatible microcontroller, an Arduino servo shield, a step-down converter, and several other customised components. The servos enable the opening and closing of the gripper as well as the manoeuvring of the arm from one side to the other, thereby exploring a 180-degree search space. The FSRs are mounted on both the inside and the outside of each of the two gripper fingers. The purpose of the FSRs on the outside of the gripper finger is to detect an object through touch during the search. The FSRs on the inside of the gripper finger are to enable detection of whether an object has been grasped or not. See Appendix A for further information.

### 2.4. Robot Task and Procedure

The robot performs a simple explore, grasp, and move task. This involves the robot moving the arm from left to right and stopping when it senses it has bumped into something, typically the target object due to changes in the FSR value readings being read continuously. If the robot does not sense an object by the end of the initial sweep, it moves slightly forward and then sweeps in the opposite direction. Once the object is sensed by the robot through a change in the FSR values being read continuously, the robot performs a vertical movement of the arm over the object, moves slightly sideways, and then descends to secure a grip on the object. The arm then closes until it reads a change in the (inside) FSR sensor readings. The (inside) FSR value change is interpreted as an object having been grasped. Subsequently, the arm lifts the object, and the default behaviour is to move to the right. After this movement, the arm descends and releases the object. If the neural net sends a go signal to the robot quickly enough, it will move to the left instead of the right and subsequently release the object (see Figure 3).

### 2.5. Cell Assemblies

The spiking neural network for the CAs was simulated in Nest [35]. The topology and interaction were specified in Python using PyNN [36]. Simulated leaky integrate-and-fire neurons were used [37] with the default parameters.

The simulation used 500 neurons with 30 synapses leaving each. Connectivity is a small world with a rich get richer topology [38]. This gives a Zipf distribution, so that some neurons have many more than 30 incoming synapses, while others have fewer. A CA with this topology is easier to ignite with less external activation than randomly or evenly distributed synapses. A 1 ms time step was used.

The complete code can be found at https://github.com/adammd1/Code-Robotic-Arm-And-Hand (accessed on 15 August 2023).

Handshaking, in this context, refers to establishing communication between the robot arm and the simulated Nest neural network. The robot runs on an Arduino board, in code specified in the Arduino language. The functionality and operation of the robot are intricately linked to the utilization of an Arduino board as its primary computing platform. The software in Arduino is tailored to the robot’s intended tasks (object detection, grasp, pick and place) and interfaces seamlessly with the hardware components and sensors. Through this integration, the robot gains the ability to perceive its surroundings, process FSR inputs, and make informed decisions based on programmed algorithms in simulated Nest and Arduino. The synergy achieved between the software and hardware components empowers the robot to carry out tasks efficiently and effectively. When the robot has grasped the object, it sends a start signal via a USB cable to the PC running the neural simulation. This starts the external activation of the neurons. When the ignition threshold is reached by the neurons, the go signal is sent to the robot arm.

## 3. Results

### 3.1. EEG

Behavioural results showed that participants were faster when responding to remembered (yes) items (mean = 852 ms, standard deviation = 577 ms) compared to new (no response) items (mean = 957 ms, standard deviation = 649 ms). There was a build-up of motor activity in the 400 ms time interval preceding the action onset for both conditions—that is, both when a response was associated with a previously remembered item (Figure 4, red line) and when a response was associated with a new item (black line). Therefore, the presence of an LRP was confirmed.

The difference in build-up of motor activity reflects the decision-making process for the item. Participants were faster to respond to items they had previously seen. There was on average a larger LRP in the 400 ms prior to action onset for the remembered items (−1.13 µV) compared to the new items (−0.49 µV).

### 3.2. Cell Assemblies

Figure 5 combines two rastergrams and spikes per ms. The blue rastergram shows when each neuron spikes with 100 neurons externally stimulated. The yellow line shows the total number of neurons that spike each ms, which is a summation of the blue rastergram. Note that the neural simulation stops recording spikes when the ignition threshold is reached; therefore, the blue rastergram stops. Similarly, the purple rastergram is aligned with the blue line and reflects only 50 neurons being stimulated initially. This shows that the firing rate, unsurprisingly, grows more rapidly with the initial external stimulation of 100 neurons than with 50 neurons.

The decision is made when there have been 1100 spikes in the last 10 ms. So, the decision is made, and the robot goes left when 100 neurons are externally stimulated. When only 50 neurons are externally stimulated, the firing in the assembly grows more slowly. The decision has already been made for the robot to go right when the signal to go left arrives. Figure 6 shows the ignition time by the number of neurons externally activated. The purple average line shows the average of 10 CAs with between 100 and 50 externally activated. The green line shows the CA from Figure 5, which passes the ignition threshold at 56 ms when 100 neurons are externally activated, and 102 ms when 50 are externally activated. All 10 randomly generated CAs behave similarly. There is a variance between different CAs, but the time to ignition quite consistently decreases as more neurons are externally stimulated.

The overall performance of the systems was largely correct. When an object was placed in the field, it was grasped over 90% of the time; most failures resulted from not grasping the object, though the arm still moved. The times were roughly 2 s from grasping to being completely lifted and about 3 s from grasping to dropping (either right or left). Videos of both versions of the task (left and right) are available on https://youtu.be/n5eMTub8MbU and https://youtu.be/-mW3yW83TGg (accessed on 13 October 2023).

## 4. Discussion

The three components of this study are the human EEG data, the robot, and a spiking neural network, and the study as a whole provides novel evidence of successfully integrating these three areas. The robot was developed to successfully perform a simple pick-and-place task. The recorded EEG data from humans provided information as to how the brain responds to a simple action. The data showed that, prior to an action, there is a build-up of neural activity, known as the LRP. This gradual build-up informed how an artificial neural network could be integrated with the robot, resulting in a cell assembly-based decision-making process being included. The firing pattern of the cell assemblies that was observed approximated the human neural data and was able to handshake successfully with the robot. It is important to note that while each component of this study is not in itself complex, together they form a platform whereby robots that are more efficient can interact with the environment. This study shows a brain-inspired robot driven by a spiking neural network.

The human brain activity data in this study are consistent with the literature as an LRP in the 400 ms time interval preceding an action [34] was observed. Moreover, the magnitude of the LRP was different depending on the input. To be more specific, when participants in the task responded yes—that they had seen an image before—the negative LRP amplitude was larger compared to when they responded no—that they had not seen the image before. This is consistent with the literature that shows that the build-up of motor activity, the readiness potential, can be influenced by the environment—variables such as amount of advance information and task difficulty [39,40,41]. It has been suggested that the increased neural firing reflects the accumulation of evidence that, when reaching a threshold, triggers action performance [42].

The robot functions as a means to integrate the neural network with the real world; it can be used to embody the brain [43]. Not to overstate the case, the body in this study is very simple, and the brain is extremely simple. The brain is one CA. When it is stimulated, there is a build-up in activity and an increase in firing in the neurons of the CA. This is largely consistent with the reported human EEG data; however, see below. In the main, it is stimulated by the activation of either 50 neurons or 100 neurons. If 100 neurons are activated, the robot receives a signal from the “brain” quickly enough to override the default behaviour. If 50 neurons are activated, the robot does not receive the signal quickly enough and performs the default behaviour. This pattern of results is in line with observations from LRP research whereby two motor commands are initiated, such as in a start- and stop-signal task [20,41]. The motor activity from these two competing actions is different. As outlined in the introduction, this has been explained by the independent horse-race model, whereby whatever motor command finishes first wins, and that action is executed [24]. The two CAs firing in the current study (see Figure 5) represent two motor commands, and the one which finishes first is executed. While this study uses a static structure, the topological structure is generated randomly, and other structures (such as all connectivity) will not allow a gradual build-up of neural firing. Other topologies (perhaps a Watts–Strogatz model [44]) may also duplicate the EEG data.

A discrepancy between the CA firing pattern and the LRP is the overall timing. The build-up of motor activity typically starts at around 400 ms pre-movement onset, whilst the CA time frame is concentrated to a 100 ms time interval. This might be improved by using a larger CA, which would take longer to ignite, or having several CAs in competition via inhibitory synapses or neurons.

Similar shortcomings in the study arise from the relative simplicity of the robot and the neural network. Indeed, one obvious way to improve the overall system would be to allow the neural network to control more of the behaviour of the robot. While there is a discernible gap between real brain recordings and the simulated neural network, it is important to note that this represents a simplified model of the task.

## 5. Conclusions and Future Directions

In conclusion, the findings from this neurorobotics study offer valuable insights into the intricate nature of human action preparation and its potential applications in the development of robotic systems. The present study adds to the growing development of brain-inspired intelligent robotics. This proof-of-concept study shows that integrating understanding of human actions with the firing pattern of CAs can be used to drive a robot. The observed lateralized readiness potential (LRP) in human participants demonstrated the gradual process of response preparation, highlighting the complexity of human motor activities. By comparing the human neurophysiological data with a robotic arm executing a pick-and-place task, the overall system successfully demonstrates the integration of human insights into machine learning. The synergy between human EEG data and the simulated spiking neural network used to control the robot arm has led to a notable alignment between human and robotic behaviour. This work emphasizes the potential for practical applications in robotics through interdisciplinary research at the intersection of neuroscience and robotics.

Future research can improve on the findings of this study by more closely matching the action that a robot performs with the recording of EEG data from humans—that is, making both human and robot perform the same task, which allows for a more precise analysis of events. The initiation of events and any changes due to sensory input to the robot can then more closely follow the timings in the human brain and mimic the relationship between activation and inhibition. Moreover, future research may aim to employ the spiking neural network to drive the full range of actions that the robot performs to create a more dynamic robot able to adapt to the environment. Although the current research is not complex in terms of robotics, spiking neural networks, or human brain recordings, in bringing these components together this project represents a significant step forward in the pursuit of human-inspired robotic systems, offering a glimpse into a future of intelligent technology attuned to the intricacies of human behaviour.

Moreover, the human brain benefits from the ability to integrate information from multiple brain regions. The state and decision making for one brain process will interact with other actions, thoughts, and decisions. Therefore, an exciting development is not only how robots can more dynamically interact with the environment, but also how they can integrate different brain processes. As Qiao, Wu, Zhong, Yin, and Chen ([45], p. 1) recently concluded “research on human-inspired intelligent robots by mimicking their biological structure, behavioural features, intelligent principles, and control mechanisms will be significant in developing new-generation robots.” This could also mean integrating cell assemblies to drive robotic actions with principles from quantum mechanics, thereby introducing a highly complex and speculative area of research that merges concepts from neuroscience, robotics, and quantum physics. Quantum principles might play a role in developing advanced brain–robot interfaces that bridge the gap between neural activity and robotic actions [46].

## Figures and Tables

**Figure 1 sensors-23-08880-f001:**
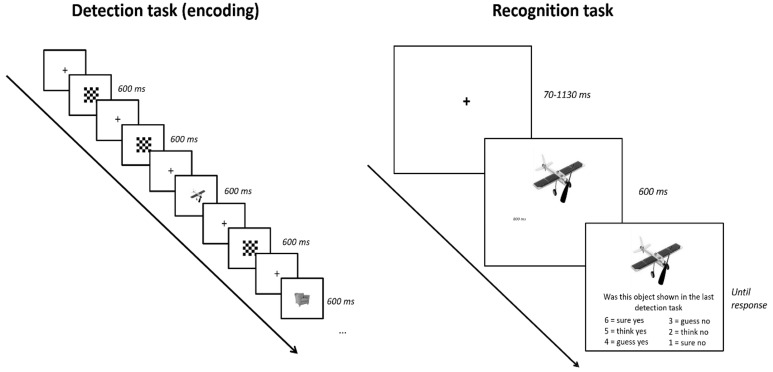
Left: Events in the detection task (not analyzed here). Objects and checkerboards were presented for a fixed duration of 600 ms and separated by a fixation cross presented for 600 ms or a variable interval (70–1130 ms) depending on condition. The participants’ task was to press the space bar if the object was an animal. Animals were infrequent targets just to ensure participants were paying attention and were not analyzed. Right: Events in the recognition task. Each object (old or new) was presented for 800 ms, after which time the participant was prompted to make a recognition judgement. If the participant had seen the object before (during the detection task) they responded yes with their right hand. If they had not seen the object, they responded no using their left hand. This yes/no decision is what the current study analyzed.

**Figure 2 sensors-23-08880-f002:**
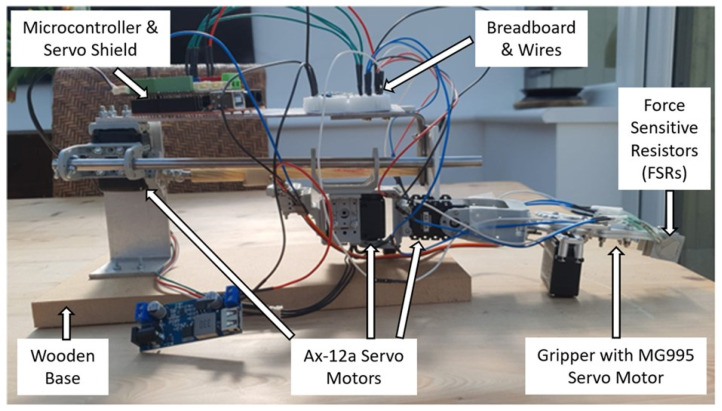
Physical robot structure for performing a task involving exploratory arm movements to locate and grasp an object using continuous FSR sensor readings. Upon successful grasp, the robot follows predefined movements before releasing the object, with a potential leftward movement based on a neural net signal.

**Figure 3 sensors-23-08880-f003:**
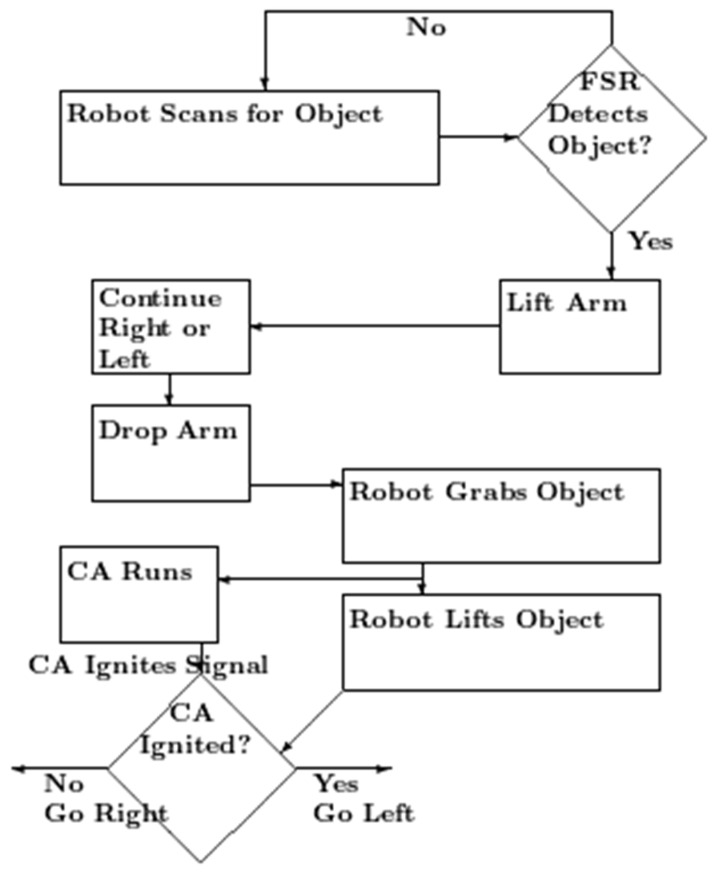
Flow chart of the robot action. The robot scans for the object, the arm moving back and forth, and forward if necessary. When the external FSR detects it, the arm moves up, and then over, so that is over the object. Then, it moves down to place the gripper around the object. The robot then grasps the object and sends a signal to start the cell assembly. It lifts the object. If the robot receives the CA ignited signal before it has started to move right, it moves left.

**Figure 4 sensors-23-08880-f004:**
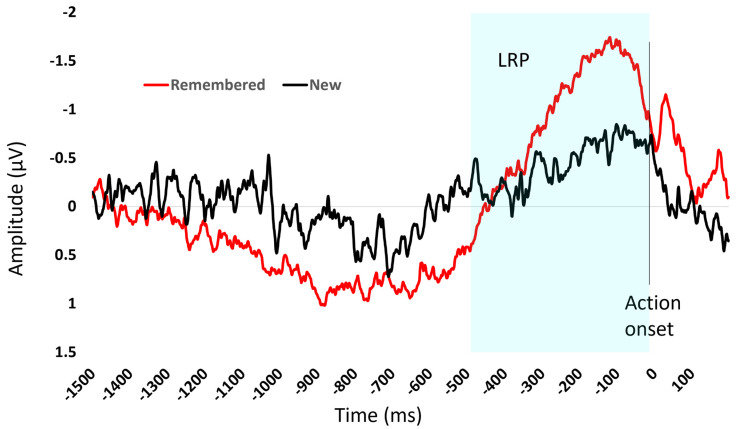
The figure shows the average event-related potential for items remembered (red) and new items (black) during the recognition task. The *x*-axis is time and 0 ms is the onset of the response. The LRP, highlighted in the blue shaded area, is the negative deflection of amplitude showing the build-up of motor activity in the time interval before action onset.

**Figure 5 sensors-23-08880-f005:**
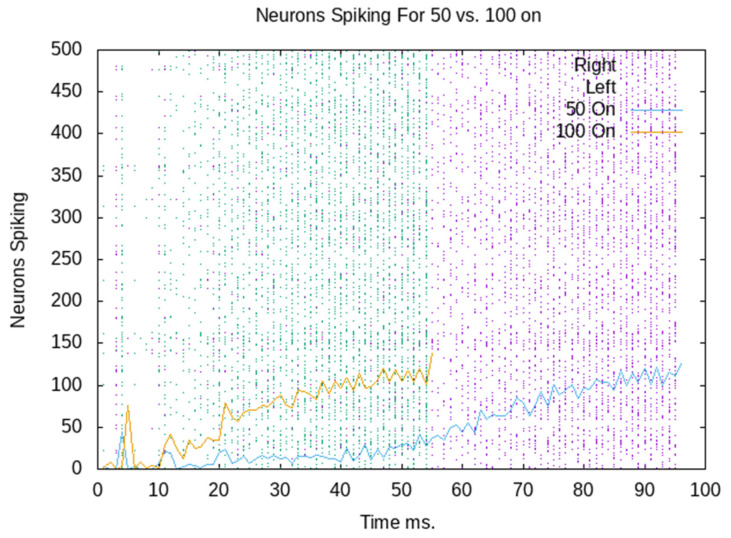
Rastergram and spikes per second of a CA ignited with 50 and 100 neurons externally activated.

**Figure 6 sensors-23-08880-f006:**
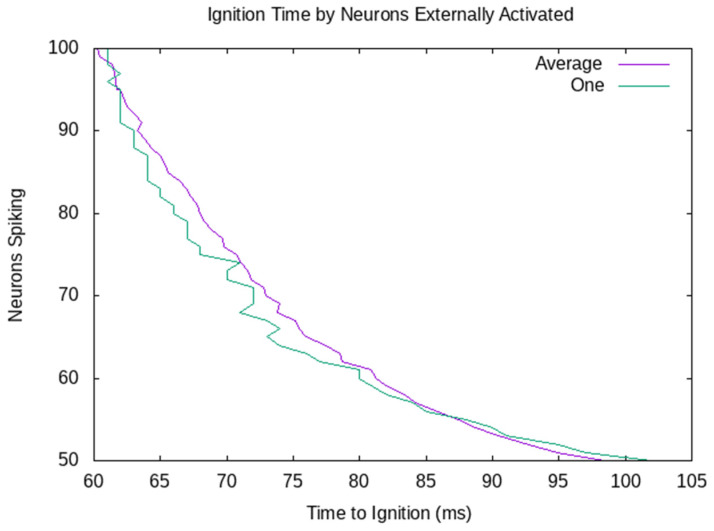
Time until ignition when 100 to 50 neurons are externally stimulated. The time to ignition decreases as more neurons are externally stimulated. Unsurprisingly, the ignition time change is more regular in the average case.

## Data Availability

Data associated with the human recorded EEG section of this study is available at: https://osf.io/tjp5u/ (accessed on 15 August 2023), Code associated with the robot task is available at: https://github.com/adammd1/Code-Robotic-Arm-And-Hand (accessed on 15 August 2023).

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
