# Peer review of "Bridging Neuroscience and Robotics: Spiking Neural Networks in Action"

_sensors, 2023, doi:10.3390/s23218880_

Round 1

Reviewer 1 Report

Comments and Suggestions for Authors

This article combines EEG signals, robotics, and a spiking neural network to conduct this research. The spiking neural network is used to simulate the function of the human brain to control the movement of a robotic arm. The work is interesting, and the results show that the neural data simulated by the robot is generally consistent with human data. However, there are still some possibilities to improve the article.

1. A flowchart of the whole system should be given in the article, which will make it easier for the reader to understand the whole principle of the research.

2. The three parts of the research described in the paper (EEG, robotic arms and pulsed neuron networks) are described relatively independently, and it should be made clear how these three parts relate to each other.

3. Since the author has designed the impulse neural network to control the robotic arm, the actual control results should be given in the paper.

4. on the left side of Fig. 1 (during the coding process), the author mentions that there are pictures of animals, it is better to give the pictures with animals in it.

Reviewer 2 Report

Comments and Suggestions for Authors

The paper provides a proof of concept study to demonstrate the benefit of the integration between neuroscience, robotics, and neural networks. The brain principle used as an inspiration in this study is the preparatory signals in premotor areas that can be a predictive of the upcoming movement task performance.

The target cognitive function brain areas, and neural response properties used in this study are well reasoned and situated to examine possible benefits of neuroscience and robotics integration. The human EEG and behavioral data are well designed and insightful about possible link between neural responses and behavioral outcomes. However, although the study claims that the robot simulation is largely consistent with the brain data, the procedure by which brain responses from EEG recordings are integrated into robots is a highly simplified description of the data and so the connection between actual brain recordings and the simulated neural network is very weak. The difference between the human cognitive task and the robot movement task is also not well justified. 

Moreover, the presented work lacks comparison studies across possible other solutions, particularly the neural network solutions that do not use this aspect of brain processing for movement control and execution. The study also lacks reporting the task performance of the robot, and instead only presents the internal or output variables of the simulated neural network. The spiking neural network architecture used in this study uses a preset structure and model and thus will not provide an insight on how different designs or parameter settings may affect the performance of the simulator and more importantly the robot behavior.

The public code availability is useful.

Comments:

It is at least expected that the revised manuscript incorporates these caveats in the discussion section of the paper and proposes alternatives that can be investigated in follow on or other studies.

Captions for figure 3 and 4 are misplaced.

In figure 4 the legend colors for 110 On and 50 On are switched. The other two legends Left and Right are not defined.

Figure 5 caption is incomplete. 

Reviewer 3 Report

Comments and Suggestions for Authors

The paper provides an example of how to integrate human brain recordings with simulated neural networks in order to drive a robot. The study is novel and showed interest to readers. However, the quality of the paper is very low. The figures are of low quality. The formatting should be improved as well.  The results provided is not inclusive. We readers can hardly understand how  robot arm executing a pick and place task was performed. How does the SNN inform the robot to place the object left or right? The conclusion part is not written like a conclusion at all.

Round 2

Reviewer 1 Report

Comments and Suggestions for Authors

This article does an interesting job, and the results show that the neural data simulated by the robot is generally consistent with human data.

Reviewer 3 Report

Comments and Suggestions for Authors

Revision is fine. Accepted.